# Design of a Low-Cost Measurement Module for the Acquisition of Analogue Voltage Signals

Sebastian Glowinski [1,2,*], Sebastian Pecolt [3], Andrzej Błażejewski [3] and Michał Sobieraj [3]

1    The State Higher School of Vocational Education in Koszalin, Lesna 1, 75582 Koszalin, Poland
2    Slupsk Pomeranian Academy, Institute of Health Sciences, Westerplatte 64, 76200 Slupsk, Poland
3    Department of Mechanical Engineering, Koszalin University of Technology, Sniadeckich 2, 75453 Koszalin, Poland
*    Correspondence: sebastian.glowinski@tu.koszalin.pl; Tel.: +489-43-478-395

**Abstract:** The aim of this work was to design and program a low-cost universal multichannel measurement card from scratch. The constructed device has analog inputs with the possibility of using them as differential inputs. This makes it possible to measure the analog signals for most of the available sensors. Thus, universality of the device is achieved. Simultaneously, the main assumption of the project and its novelty was to develop a measurement module. It is characterized by high measurement parameters, comparable to commercial products available on the market, with a very low production cost. The usability and assumed features of the measurement module were verified and tested using a functional generator and constructed test stand. During the tests, a sampling rate of at least 250 kS/s and a resolution of at least 14 bit were used. The module enables the acquisition of analog signals with voltages in the range of $\pm10$ V and digital signals in the transistor–transistor logic (TTL) 5 V standard with a frequency of at least 250 kS/s. In addition, our device can be controlled via a computer, and data can be downloaded via the USB interface. It has 16 input channels with the possibility of differential measurements. The proposed solution is several times cheaper than commercial solutions while maintaining comparable parameters, as shown in the conclusion of the work.

**Keywords:** analog–digital converter; field programmable gate array; microcontroller unit; multiplexer

## 1. Introduction

Electronic laboratories have many measuring devices, such as multimeters, oscilloscopes, logic analyzers, and spectrum analyzers, as well as devices that generate signals, such as function generators and programmable power supplies [1,2]. These devices are useful and are sometimes necessary for designing electronic circuits and solving related problems [3]. The data acquisition modules are multitasking devices [4]. They are used to conduct research or testing in an automated manner. Module capabilities differ depending on the manufacturer and model. The most common module capabilities are voltage measurements using an ADC, digital inputs capable of reading logic states with TTL levels, and digital outputs that allow digital signals to be generated with TTL levels. Additional functions are often analog outputs, owing to which the module can be used in control systems or can generate voltage signal waveforms that act as a force on the tested object. A great advantage of these modules is their multichannel nature [5]. Modules are usually capable of acquiring data with many signals simultaneously, which requires the use of many such devices in the case of typical measuring devices [6]. In addition, they allow the use of a single device to test multi-signal and complex processes or phenomena. However, a disadvantage of these devices is their high costs. These measurement modules can be used for exoskeletons sensors and prostheses [7,8]. The increased amount of data obtained from these types of devices requires specialized measurement tools. In the past several decades, remarkable progress has been made in the development of exoskeletons [9].

For example, they can monitor the pressure and temperature at the skin-prosthesis interface. This can be valuable in the fitting process and in monitoring the development of dangerous regions of increased pressure and temperature as limb volume changes during daily activities [10]. In embedded systems, users receive data based on electrocardiogram (ECG), electromyogram (EMG), or electroneurogram (ENG) signals [11]. The proposed platform enables the fusion of various sensor data with the objective of motion identification and prosthesis control by reading out various data (forces, acceleration, data, etc.) and integrating identification algorithms.

A different approach to the problem of measurement cards was presented in [12,13]. However, as a new approach, the authors used the criterion of the lowest possible costs in relation to the parameters. Data acquisition modules are produced by many manufacturers, but they are expensive. The cost of data acquisition from recognized producers can be as high as several thousand dollars. The hardware and firmware design of a low-cost computer-based data acquisition system based on the open-source philosophy was described by Ferrero Martin et al. [14]. The proposed system was interesting for conducting laboratory experiments and industrial applications.

A single-chip solution that allows a wide variety of operations required by sensor systems, such as vector impedance, voltage, and current measurements across a wide frequency range, was presented by Ria et al. [15].

A novelty in this area is the presentation of the design process of a functional measurement card, built-in non-commercial (non-laboratory) conditions, with ready-made electronic elements, and with the open Unix system, which can compete with commercial products. The constructed device has analog inputs with the possibility of using them as differential inputs. This makes it possible to measure the analog signals for most of the available sensors. Thus, the universality of the device is achieved by the open system with the possibility of its configuration and further extension.

The first part of this paper focuses on the selection of the topology system for the analog part. Next, the authors present a schematic, PCB design, and measurement module software. In the next section, the correctness of the system operation is verified, and the results are presented. Finally, a brief discussion and the limitations of the study are presented.

## 2. Materials and Methods

During the design of the system topology, it was found that the multiplexed topology was almost three times cheaper than the nonmultiplexed topology. In addition, the non-multiplexed solution takes up more than twice the area of the PCB. The non-multiplexed topology allows for the simultaneous operation of all channels, with an almost full sampling rate, but it is associated with the requirement to ensure greater data throughput. Figure 1 shows a measurement station and the designed PCB measurement module and the dimensions.

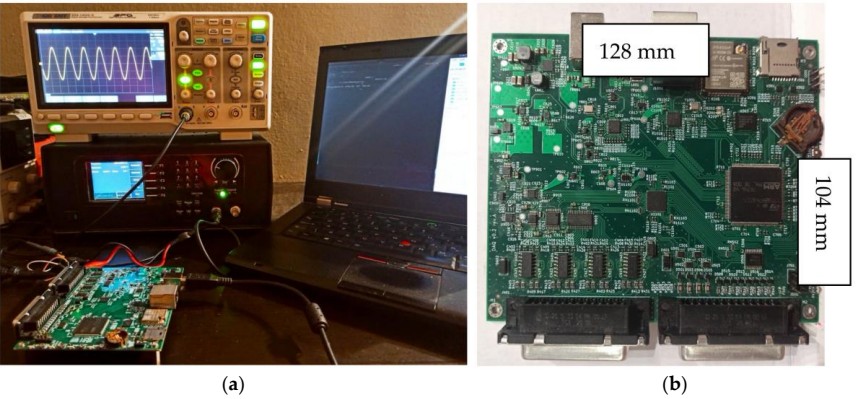

(a)                    (b)

**Figure 1.** Measurement station (**a**) and the designed PCB measurement module (**b**).

The measuring station was designed, assembled, and programmed (Figure 1a). Its performance was verified and tested using a proprietary functional generator. The generator is based on the Spartan 6 XC6SLX9 FPGA chip that controls the AD9117 D/A converter and the STM32F407 microcontroller that controls the FPGA and supports the user interface [16]. A Siglent SDS1202X-E oscilloscope was used [17]. The oscilloscope has two channels with a 200 MHz band, sampled with a converter with a sampling frequency of 1 GS/s. An oscilloscope was used to verify generator settings. Communication with the measuring module was achieved by using a laptop. The measurement results obtained in the form of CSV files were entered into Octave software to plot graphs [18]. Figure 1b shows the finished PCB of the developed measuring module.

### 2.1. Selection of the Topology of the Analog Part

The device design begins with the part responsible for shaping and sampling the analog. The interface through which it communicates with the digital component depends on the solution. The number and type of circuits used in the analog part depend on the data capacity that must be achieved by the communication interface with the device, as well as the number and types of buses connecting the digital part with the analog part. The converter that meets the design requirements is an AC AD7942 from an analog device [19]. It has a resolution of 14 bit and a sampling frequency of 250 kS/s. It includes pseudo-differential inputs, meaning that it has inverting and non-inverting inputs, of which the inverting input must be held at a potential equal to the circuit ground. This allows it to be connected directly to the ground of the signal source, eliminating measurement errors arising from potential differences between the ground of the converter and the source of the signal processed by it [20].

The design assumes the possibility of using inputs in a differential configuration, so that each converter in this configuration would have to receive the signal processed by the differential amplifier. The input signal should be buffered so that the system tested by the device is not loaded with low impedance. In addition, the signal would have to be suppressed, as the converter will be damaged by voltages greater than its supply, which for this converter is typically 5 V. Therefore, it should be assumed that a measuring amplifier is used to ensure high input impedance of the measuring system and simultaneously enable differential measurements (Figure 2). The instrument amplifier diagram shown in Figure 2a shows typical signal shaping and the Vref designation refers to the common ground. Assuming 16 input channels, it is possible to simultaneously sample all the channels at 220 kS/s using two SPI interfaces, one per eight daisy-chained converters. The interface for communication with the computer must, therefore, ensure a data throughput of not less than 6.7 MB/s, assuming that each 14-bit sample will be sent in the form of two bytes. The block diagrams of the solution are presented in Figure 2b,c.

Multiplexers were used to switch the channels sampled by a single ADC. In this topology, the only duplicate circuit is the input buffer, which provides high input impedance. The ability to study differential signals can be achieved using two multiplexers and a pair of signals for differential measurement. Using the proposed multiplexer, it was possible to combine 32 inputs into 16 differential inputs. During the design of the system topology, it was found that the multiplexed topology is almost three times cheaper than the non-multiplexed topology. Additionally, the non-multiplexed solution takes up more than twice the area of the PCB. The non-multiplexed topology allows for the simultaneous operation of all channels, with an almost full sampling rate, but it is associated with the requirement to ensure greater data throughput Table 1.

Based on the available solutions, it can be concluded that the multiplexed solution is almost three times cheaper than the nonmultiplexed solution. Input multiplexing meets the design requirements while reducing the design cost and dimensions of the analog part of the device; therefore, this topology was used in the design.

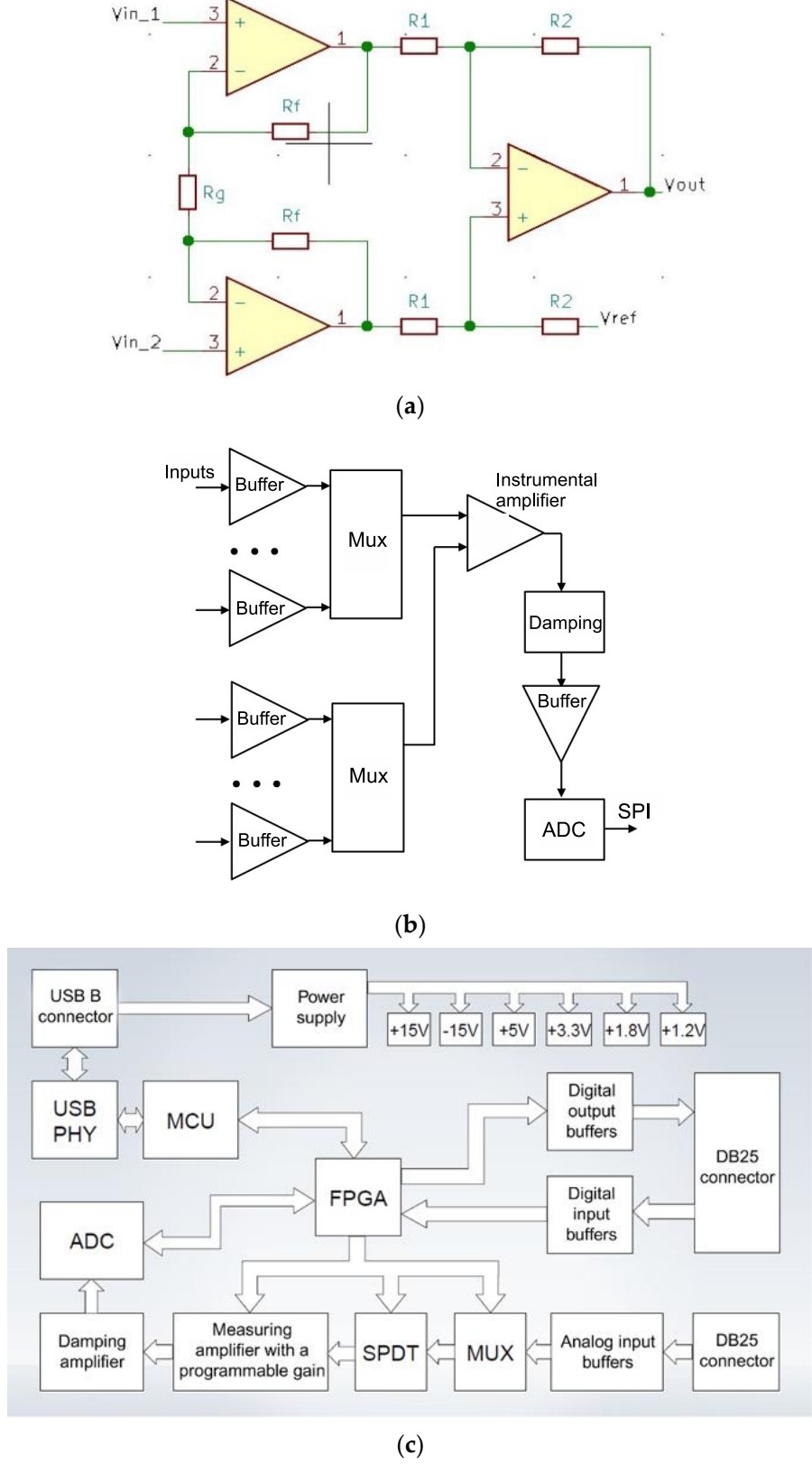

**Figure 2.** Instrumental amplifier (**a**), a block diagram of the designed measuring module together with the direction of the flow of signals from individual subsystems (**b**), and a block diagram of the device (**c**).

**Table 1.** Summary of the results of the topology analysis of the analog part of the project.

| Topology | Cost [PLN] | Occupied Area [cm$^2$] | Maximal Frequency when Sampling a Single Channel [kS/s] | Maximal Frequency when Sampling Measurements of All Channels [kS/s] | Summarized Frequency when Sampling Measurements of All Channels [kS/s] | Required Capacity Data [MB/s] |
|---|---|---|---|---|---|---|
| Not multiplexed | 608 | 7 | 250 | 220 | 3520 | 6.7 |
| Multiplexed | 226 | 3 | 250 | 15,625 | 250 | 0.5 |

*2.2. Selection of Components for the Analog Part*

OPA4197 was used as the input buffer [21]. The inputs of this system were filtered against EMI and RFI, which reduced the measured amount of interference that reached the device through the test leads. In addition, the output of the amplifier has a high maximum output current of up to 65 mA and can work with capacitive loads of up to 1 nF, which is important because the buffer outputs are connected to the multiplexer inputs, which load the buffers capacitively during switching. The solution proposed during topology selection guarantees 16 channels with differential inputs. However, the signals are often not measured differentially; therefore, it is worth considering the possibility of configuring the inputs and measuring every single-voltage signal related to a common ground. The input configuration can be provided by using two analog SPDT switches. The outputs of both multiplexers are connected to one of the switches, and the outputs of the second multiplexer and ground are connected to the other switch (Figure 3a). In such a configuration, it is possible to pass any of the multiplexer outputs to the output of the first switch, while maintaining the output of the second switch at the ground potential. When there is a need to perform a differential measurement, the first switch selects the output of the first multiplexer, and the second switch selects the output of the second multiplexer; thus, the switch outputs a pair of differential signals made up of the outputs of both multiplexers. Because of this solution, it is possible to use multiplexers with a smaller number of inputs while maintaining the same number of device inputs. However, the number of differential inputs was twice as small as that in the ground reference configuration. The applied switch should ensure that the switching time is not greater than that of the multiplexer. Moreover, it must have the lowest possible switching load because its inputs are not buffered, but are directly connected to the outputs of multiplexers, which do not have such a high output current as the input buffers. A TMUX6136 chip from Texas Instruments was selected, which contained two analog SPDT switches in one housing. The circuit has a switching time of 66 ns, which is significantly lower than that of the mux used in the exemplary representation of the multiplexed topology, where the switch time was 400 ns. The switching charge is ten times lower than this parameter in the exemplary multiplexers, being 0.4 pC (Figure 3b).

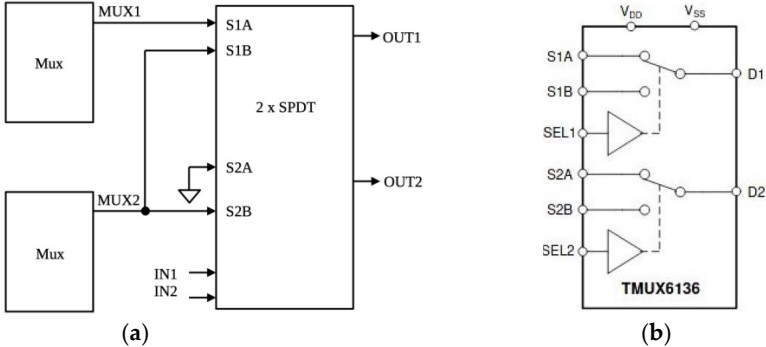

(**a**)                          (**b**)

**Figure 3.** Configuring differential inputs using analog switches (**a**); simplified schematic of TMUX6136 (**b**).

By using analog switches to configure the differential inputs, it is possible to maintain 16 input channels by using two 8-channel multiplexers instead of the 16-channel multiplexers used in the example during topology selection. To achieve the lowest possible time to determine the entire signal-shaping path, it was decided to choose a multiplexer with a lower switching load, owing to which the input will load the input buffers less when changing the channel. A Texas Instruments MUX36D08 chip was used in this study [22]. It comprises two 8-channel multiplexers that are controlled by the same control signals. Owing to the use of this system, space on the printed circuit board is saved, and switching control is also facilitated, as both multiplexers are controlled with the same signals. Moreover, configuring differential inputs always results in the same pairs of signals. The multiplexer has a switching time of 92 ns, which is more than four times shorter than in the example solution, which is not necessary, but is associated with a much smaller switching load, which amounts to 0.31 pC, i.e., over ten times less than in the ADG506 system.

It was assumed that the measuring amplifier would be created as a system of three separate amplifiers; however, such a solution has a significant disadvantage, which is the low common-mode rejection coefficient. This results from the use of the external passive components available in the market. Considering the circuit of the differential amplifier, ignoring the frequency domain, it can be seen that the tolerance of the resistors used in the circuit has a significant effect on the *CMRR* of the circuit. The predicted common mode rejection factor for a DC signal source is calculated as follows:

$$CMRR \approx 20 \log \left( \frac{\frac{1}{2}(G+1)}{\frac{\Delta R}{R}} \right), dB; \tag{1}$$

where *CMRR* is the common mode rejection coefficient related to the tolerance of the resistors used, *dB*; *G* is the gain of the differential amplifier, V/V; and Δ*R*/*R* is the resistor selection ratio resulting from their tolerance.

Using this formula, it can be determined that the predicted *CMRR* using resistors with a tolerance of 1%, that is, the worst-case selection ratio will be 2%, and with a gain of 1 it will be approximately 34 dB. For resistors with a tolerance of 0.1%, the coefficient was approximately 54 dB. The achieved *CMRR* of 54 dB is not satisfactory; however, assuming a 14-bit converter and measuring signals in the range of ±10 V, the least significant bit of the converter will correspond to 1.2 mV, with the common mode attenuation coefficient being 54 dB, and 1 V of the common signal corresponds to approx. 2 mV of the measurement error, which is greater than the voltage equivalent of 1 LSB of the transducer, and is not acceptable. Therefore, a solution with a higher *CMRR* should be used.

Integrated circuits containing complete amplifiers in a single housing often have a high *CMRR* because of the production matching of components in the integrated circuit itself. However, most commercially available off-the-shelf instrumentation amplifiers are unsuitable for this project because they do not have an appropriate output voltage ramp to keep up with channel switching.

An AD8250 chip was used as the instrumentation amplifier [23]. This is one of the few measurement amplifiers suitable for use in multiplexed acquisition systems with switching frequencies greater than 250 kHz. The system ensured stabilization of the output voltage to 0.001% in approximately 600 ns. In the case of a 14-bit converter, stabilization of up to 0.006% was sufficient. The amplifier has a *CMRR* of 98 dB, which means an error of approximately 13 µV in the 1 V common mode. This is much lower than the value required to operate a 14-bit converter. In addition, it is an amplifier with programmable gain, with possible gains of 1, 2, 5, and 10. Therefore, it is possible to configure the channel for a smaller voltage range while maintaining the high resolution of the converter [24].

An AD8475 chip was used as a damping amplifier. It is a fully differential damping amplifier that allows constant attenuation through gains of 0.8 or 0.4, depending on the configuration of the inputs. The amplifier is capable of a steady state down to 0.001% in 50 ns when changing the output by 2 V; thus, it is sufficiently fast to sample at a given

frequency. In addition, a voltage, which is a common output signal, can be introduced into the system. The system provides a range of input signals of 15 V with a single 5 V voltage supply, which ensures the safe conversion of signals without the need to use additional protection at the ADC input, because the AD8475 chip will not be able to exceed the permissible input voltage of the converter [25]. The amplifier was used in a configuration with a gain of 0.4 and a voltage of 2.5 V was introduced into the output signal. Therefore, using an ADC converter with a differential input and allowed input voltages of up to 5 V, signals in the range of 10 V are converted into a pair of differential signals, the voltages of which range from 0.5 to 4.5 V. This allows for an increase in the effective measuring range to 12.5 V, at which the differential signals reach the range from 0 to 5 V. The assumption of the scaling configuration was simulated using LTSpice [26]. The AD8475 system was powered with 5 V, the inverting input with a gain of 0.4 was connected to the ground, and a sinusoidal voltage with an amplitude of 12.5 V and a frequency of 1 kHz was introduced to the second input. A common signal with a voltage of 2.5 V was introduced into the system (Figure 4). The reference voltage source REF5050 was used as the reference voltage for the ADC. This circuit has the highest parameters in terms of stability, so it was also used for VOCM. The 2.5 V voltage was realized by a simple voltage divider from high tolerance resistors 2 x 10 k ohms and a voltage follower on the AD8538. This solution allowed the use of one high-end reference voltage source and ensured the stability of the VOCM.

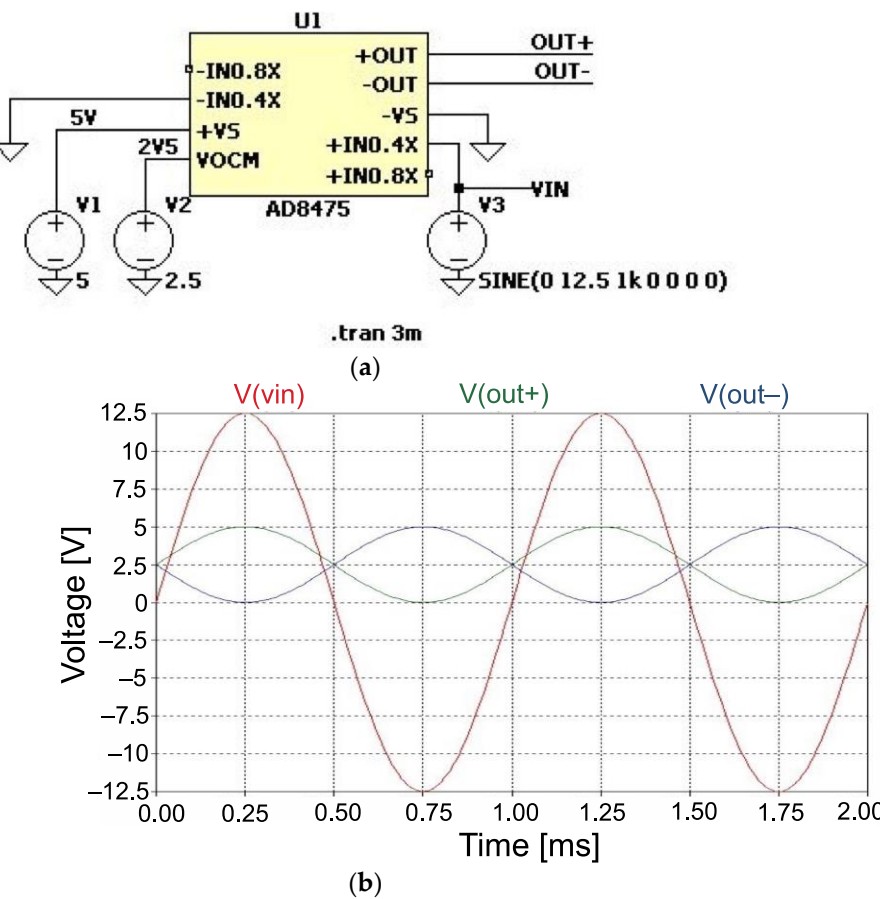

**Figure 4.** Scaling test diagram in LTSpice (**a**). Scaling simulation results using AD8475 in LTSpice (**b**).

The project used an ADS8318 converter from Texas Instruments. It is a 16-bit ADC converter with a sampling frequency of up to 500 kS/s. The circuit has a differential input and accepts reference voltages of up to 5 V, so it can process the signals generated by the damping amplifier. The conversion time of this converter was 1400 ns and the acquisition time was 600 ns. Communication with the transducer was achieved through an interface compatible with the SPI.

### 2.3. Time to Settle the Analog Part

Settling time is the most important parameter of the analog part in a multiplexed data-acquisition topology. The system must stabilize the ADC to an accuracy of less than 1 LSB after switching the channels in less than one sampling cycle. The fixing time can be estimated using the following dependence:

$$t_{fixing} = \sqrt{\sum_{n=1}^{k} t_n^2}, s; \tag{2}$$

where $t_{fixing}$ is the time to fix the entire system, $s$; $t_n$ is the time for determining subsequent elements, $s$; and $k$ is the number of elements in the system.

In the next step, the time required to set the multiplexer and analog switches with the required precision is determined. For a 16-bit converter, the determination precision should be 0.001%. Because of the direct connection with the analog switch, the two circuits must be connected during the simulation, and the parameters of the switch should be considered. The schematic used for the simulation in LTSpice is shown in Figure 5a.

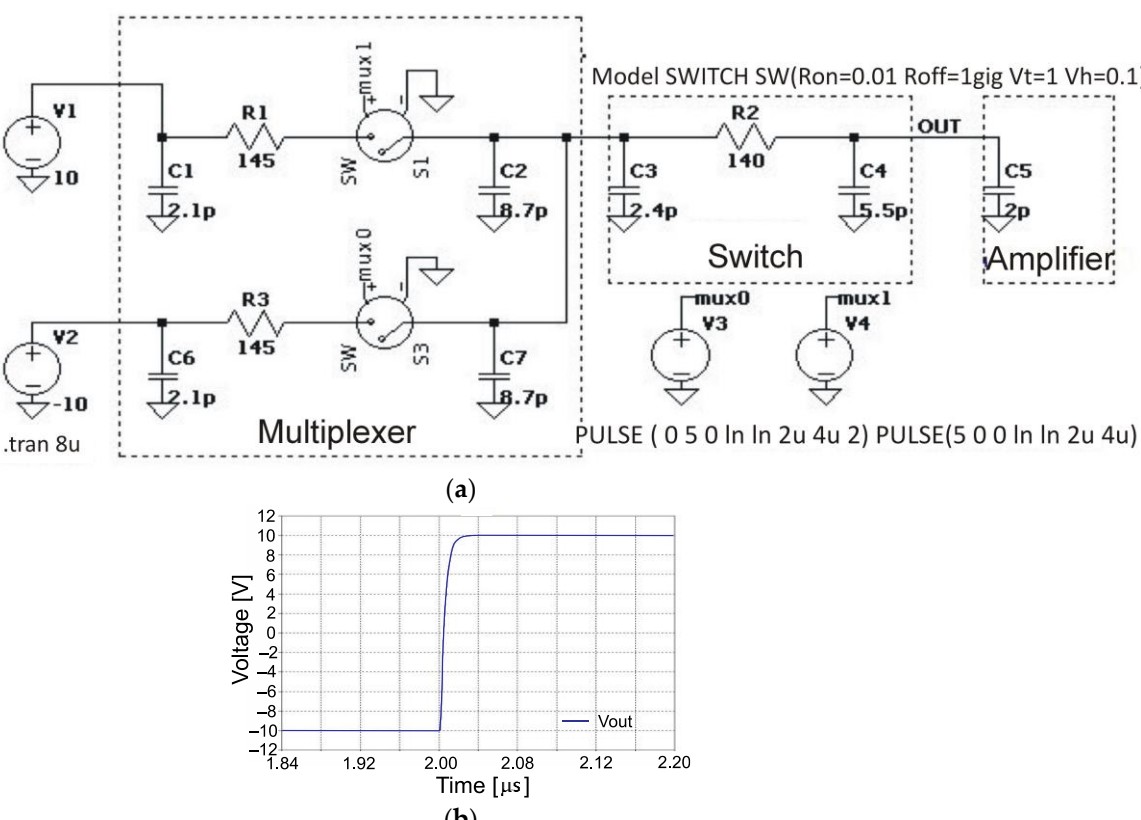

(**a**)

(**b**)

**Figure 5.** Multiplexer setting time simulation scheme (**a**); switch output setting simulated in LT-Spice (**b**).

The diagram considers the multiplexer parameters, switch parameters, and the input capacity of the measuring amplifier to which the switch outputs are connected. The resistive part of the amplifier input impedance is omitted because of its high value (1.25 GΩ). After performing the simulation for switching between the limits of the assumed measurement range of ±10 V, the settling time to 0.001% of the target value was measured to be 52 ns using the cursors available in the software (Figure 5b). The results of the multiplexer settling-time simulation were surprisingly low because the input buffer was not included in the simulation. When switching the multiplexer, it capacitively loads the output of the input buffer, resulting in a decrease in its output voltage. To prevent this, an RC network is used between the multiplexer and the buffer. The capacitor in the RC network functions

as a power bank, which is consumed by the multiplexer when the channel is switched on. However, op-amps become unstable if they are loaded with too much capacitance; therefore, the amplifiers charge the capacitor through an isolating resistor. We prepared a second simulation considering the input buffer. The selection of the value and ratio of the RC network elements was performed experimentally by iterating a simulation through pairs of values (Figure 6a).

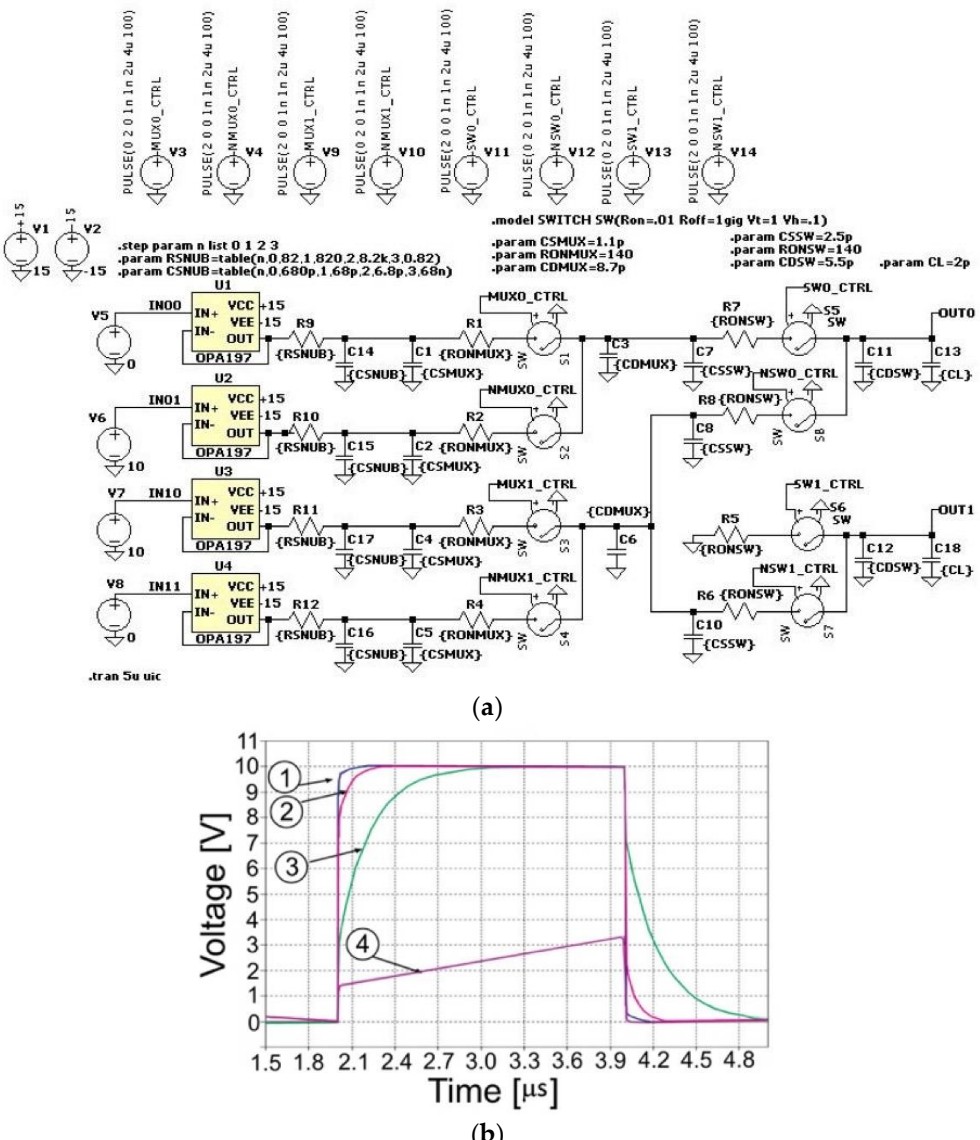

(**a**)

(**b**)

**Figure 6.** Scheme of the simulation of the settling time of the multiplexer and switch considering the input buffers (**a**). Results of the simulation of the settling time of the multiplexer and switch considering the input buffer (**b**).

To determine the settling time and simultaneously select the RC network value between the multiplexer and the input buffer, iterative simulations were performed for the RC network values shown in Table 2. After the simulation, it can be seen that option No. 1 gives the best results (Figure 6b). Option 4 gave the worst results, which can be expected because less than 1 Ω is often too small for a resistance separation from a capacitance of 68 nF for an operational amplifier. Based on the best result, a settling time of 0.001% equal to 660 ns was determined.

**Table 2.** RC network simulation iterations of input buffer.

| L.p. | R [Ω] | C [F] |
|------|-------|-------|
| 1 | 82 | 680 p |
| 2 | 820 | 68 p |
| 3 | 8.2 k | 6.8 p |
| 4 | 0.82 | 68 n |

In the next step, the time required to fix the amplifiers is determined. For the AD8250 chip, the highest settling time of 0.001% was 685 ns with a gain of 10. For the AD8475 chip, the settling time to 0.001% was 50 ns when changing the output by 2 V, so it can be estimated that for 5 V, it will be 125 ns. The sum of the settling time, using the settling time of the multiplexer and analog switches from the simulation in the version with the input buffer, and the settling time of both amplifiers was 960 ns, i.e., twice the time required because of the ADC sampling frequency, which is 2 μs.

*2.4. Selection of Digital Part Components*

The switching of channels and gains in the analog part must be carried out with strict precision in order to be able to use the maximum potential of the converter. Therefore, we decided to control the analog part using an FPGA chip to ensure sampling repeatability. The choice was based on the lattice ICE40UP5K chip [27]. It is an FPGA chip from the ICE 40 Ultra Plus family with 5280 LUT elements and 80 kb of pseudo-two-port RAM, that is, memory with two independent access ports: one for reading and the other for writing data. The pseudo-two-port memory can be used as an initial FIFO buffer for data downloaded from the ADC converter and digital inputs. In addition, the FPGA chip contains 1 Mb of single-port RAM, which can store the configuration data of subsequent samples, that is, the channel and gain selection.

An FPGA chip can also be used to collect digital samples. It is only necessary to select appropriately protected input buffers to protect the system and to ensure the possibility of accepting samples with TTL 5 V voltages. Nexperia 74LVC541 was used as the input buffer [28]. The system consists of eight digital buffers resistant to signals in the TTL 5 V standard, powered by lower voltages. The system was protected using a serial current-limiting resistor followed by control diodes that limited the voltage to 5 V. The power line to which the current is discharged in the event of an excessive voltage at the digital input is protected by a TVS diode.

An STM32H743Z chip from STMicroelectronics was used as a microcontroller. It is a microcontroller based on the Cortex-M7 core with clock speeds up to 480 MHz [29]. The system has a built-in 2 MB flash memory and 1 MB RAM memory. RAM includes several dozen of KB of TCM memory, which allows instructions to be executed without waiting cycles. It supports USB 2.0, using an external physical layer system, as well as Ethernet with a throughput of up to 100 Mb/s, also using an external PHY chip. The system has many DMA controllers that allow data transfer without overloading the processor, which is important when simultaneously downloading data from an FPGA and sending them to a computer.

For ease of use, we decided to base the data transmission on the USB 2.0. This allows the device to be connected to a computer and powered with the same cable. The microchip USB3320 was chosen as the physical layer for USB 2.0 [30]. It is a PHY chip that meets the requirements of the USB 2.0 standard and communicates with the microcontroller via the ULPI interface, supported by the selected microcontroller. This chip was chosen mainly because of its repeated use by the manufacturer of the microcontroller on the evaluation boards; therefore, it is certain that it will work smoothly with the MCU.

Most analog parts require a symmetrical power supply with a voltage of ±15 V, higher than the voltage provided by the USB interface. Therefore, a boost converter was required. However, because of the nature of the converter operation, they should not be used to supply analog circuits that are susceptible to interference. We decided to use a converter

that generates a symmetrical voltage higher than the required ±15 V and to supply LDO regulators with it, the high PSRR of which suppresses the noise generated by the converter. ADP5071 from Analog Devices was chosen as a converter capable of generating voltages of the order of 16 V from voltages as low as 5 V. ADP7142 was selected as the positive and ADP7182 as the negative voltage regulators, respectively. The regulators are characterized by a voltage drop of no more than 200 mV at 200 mA of an output current, thanks to which they will be able to maintain the output voltages at a level of ±15 V when feeding them with voltages of 1 V higher. In addition to high voltages, the analog part also needs two 5 V voltages. One of them powers the AD8475 damping amplifier and the ADC. We decided to use an LDO LDL212 regulator from STMicroelectronics [31]. The second 5 V voltage is the reference voltage for the converter, which also determines the common voltage at the output of the damping amplifier. A Texas Instruments REF5050 chip was selected, which provided voltage generation with an accuracy of 0.1% and a temperature stability of 8 ppm/°C. The digital part was powered by 3.3 V via a buck converter to ensure high power efficiency and minimize unnecessary dissipation. A TPS62063 chip from Texas Instruments was used to achieve an efficiency of up to 97%. The high switching frequency of the converter (3 MHz) allows for the use of a smaller coil. The PHY circuit of the USB interface and the FPGA circuit require 1.8 and 1.2 V, respectively, and LDO MIC5365 and MIC94345 regulators have been selected for them, which will supply the output of the converter generating 3.3 V for the digital part (Figure 7a) [32].

An FPGA or microcontroller can only be used to achieve a more compact circuit structure. However, two separate digital circuits were used to divide the tasks, significantly reducing the total load on the system. Simultaneously, this solution enables an optional module with several additional functions. The use of two ready-made digital circuits slightly increases the cost of manufacturing the module and simultaneously makes the structure of the system modular.

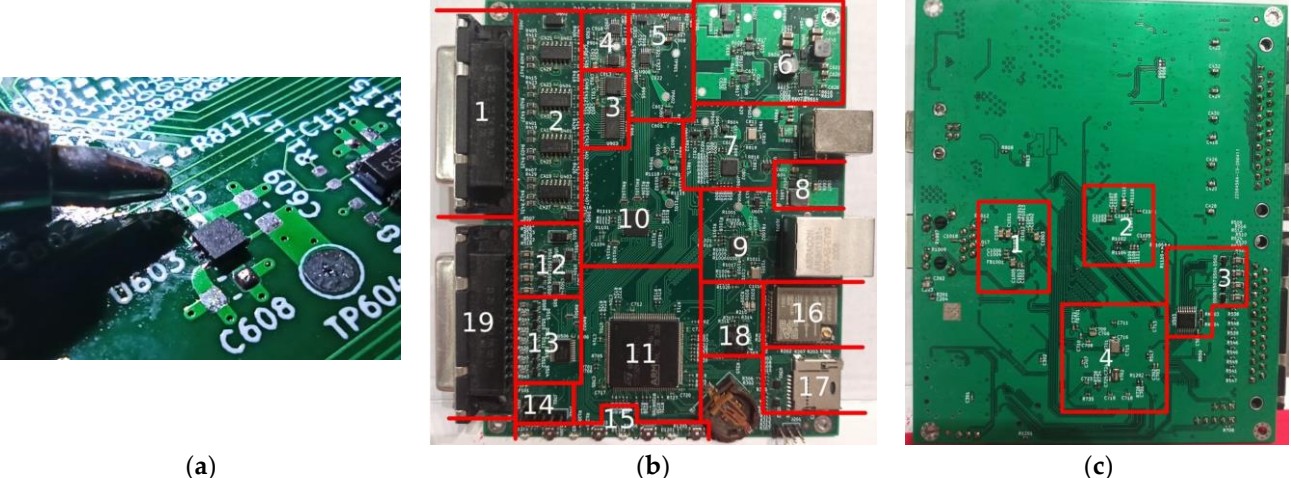

(**a**)                                                        (**b**)                                                        (**c**)

**Figure 7.** MIC94345 mounted on a PCB (**a**); PCB—top view (**b**); PCB—bottom view (**c**).

### 2.5. Schematic and PCB Design

Free KiCad software was used to create the schematic and PCB design, which allows the design of PCB projects of any size [33]. When drawing the diagram, the focus was on dividing the project into blocks to divide the entire diagram into sub-diagrams in A4 format. Using larger formats often leads to unreadable, overly complicated diagrams, where it is difficult to find connections between layouts, even when browsing them on a computer. The design included 520 components, most of which must be fitted for the device to work properly.

Tables 3 and 4 contain a list of sub-systems available on the PCB according to the numbering in Figure 7b,c, respectively.

**Table 3.** Sub-systems placed on the PCB (Figure 7b).

| Number of the Sub-System | Subsystems Placed on the PCB |
| --- | --- |
| 1 | Connector for analog signals |
| 2 | Input buffers with RC networks |
| 3 | Multiplexer and analog switches |
| 4 | Instrumental and damping amplifier |
| 5 | ADC with source reference voltage |
| 6 | Power supply for the analog part |
| 7 | USB PHY with USB type B connector and local 1.8 V regulator |
| 8 | 3.3 V converter |
| 9 | Ethernet PHY with an RJ45 connector |
| 10 | FPGA system |
| 11 | Microcontroller |
| 12 | Digital output buffers |
| 13 | Digital input buffers |
| 14 | Microcontroller programming connector |
| 15 | Additional buttons and LEDs |
| 16 | WiFi module |
| 17 | MicroSD card connector |
| 18 | Additional flash memory |
| 19 | Digital signals connector |

**Table 4.** Sub-systems placed on the PCB (Figure 7c).

| Number of the Sub-System | Subsystems Placed on the PCB |
| --- | --- |
| 1 | Passive elements of the PHY circuit of the Ethernet interface |
| 2 | Passive elements of the FPGA circuit |
| 3 | Elements of digital output buffers |
| 4 | Passive elements of the power supply and quartz resonators of the microcontroller |

*2.6. Measurement Module Software*

Basic functionality is in the form of taking analog and digital samples and then saving them to a computer requires writing three programs:

- FPGA software that allows samples to be taken from the analog part of the module and digital input buffers and then reads them from the internal memory of the system;
- MCU software that configures the FPGA chip, retrieves data from it, and enables the data to be read from a computer using the USB 2.0 interface;
- PC software that allows control of the module and downloads data from it.

The FPGA system software was created in the form of modules whose combined inputs created the entire software. Communication with FPGA was decided to be based on the SPI interface, which can achieve throughputs higher than the required 1.5 MB/s. The SPI module is based on two shift registers, one of which, together with the clock edges, latches the incoming data and the other exposes subsequent outgoing bits (Figure 8a).

The microcontroller software is based on the C language, which, despite being classified as a high-level language, allows precise control over code execution and memory usage. The software was written in the STM32CubeIDE environment, free software based on the Eclipse environment prepared by the manufacturer of the microcontroller [34]. The environment allows for easy configuration of MCU peripherals, but this option should be used only for initial commissioning and as an example of hardware service because functions in HAL libraries provided by the manufacturer are slower than manually written procedures, owing to their universality. Through this, they must execute many more conditional statements to check the situation in which their current use fits. The real-time operating system FreeRTOS was used to write the software, a free RTOS that allows the writing of multithreaded software [35]. It was decided to use a real-time operating system

because the software must constantly handle two separate tasks: FPGA interface support and USB interface support.

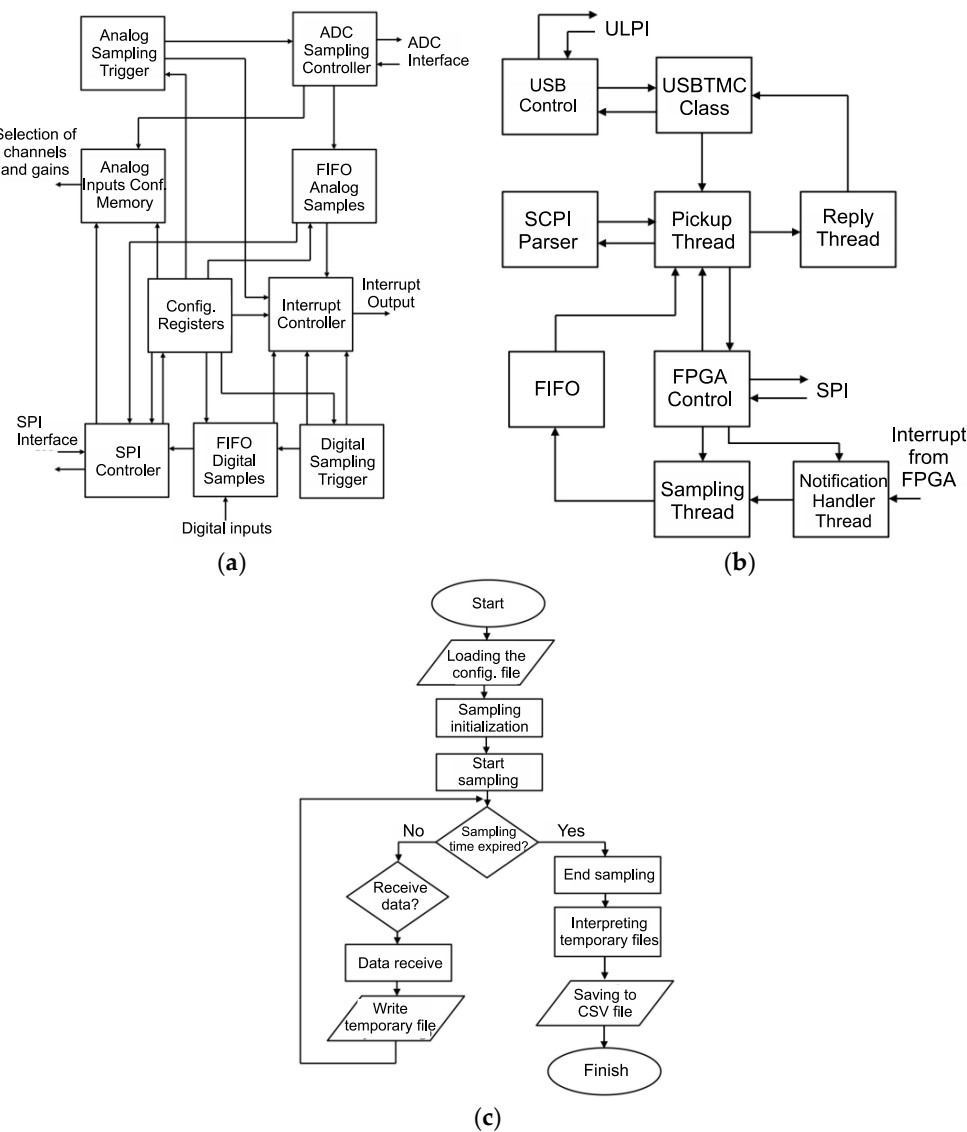

**Figure 8.** Simplified block diagram of FPGA software (**a**). Block diagram of the relationship between MCU software elements (**b**). Simplified program operation diagram on a computer (**c**).

When writing the FPGA system support, it was also necessary to write the programming procedure for the system itself, because the purchase of the programmer for this system and the output of the configuration connector were abandoned in favor of programming the system through the SPI interface in the slave mode. The manufacturer described the procedure in the documentation, which was limited to resetting the FPGA and uploading a data stream. The clock cycles were maintained until the CDONE output of the FPGA increased, indicating the completion of the configuration. Then, 48 clock cycles must be provided, which unlock the IO of the chip, after which the FPGA is configured and ready to go. To retrieve data from the system, it is necessary to implement the FIFO buffer in the microcontroller software. Therefore, a simple implementation was developed that allows the dynamic creation of a buffer of any size with elements of any size. The buffer differed from typical implementations in that it did not have the function of writing and reading the item, and instead it had only the function of incrementing the read and write

pointers, because the items had to be copied to the buffer as quickly as possible using the DMA controller; the procedure of manually copying would be too slow.

The USB interface, using the HS mode supported by the USB 2.0 standard, reaching 480 Mb/s throughput, was implemented using ready-made libraries from the manufacturer. This decision was made because it would take a long time to implement the USB support manually. The device should use a class to measure it. This class is USBTMC, which is popular in laboratory equipment such as oscilloscopes and function generators. The class is based on two data channels: input and output, in addition to the control channel. The data sent over these channels must satisfy the requirements specified in the class specification for data headers. However, this document does not specify the transfer content, implying that an additional protocol layer should be applied to this class. The class also has a USBTMC488 subclass that uses an additional interrupt channel to automate device status reporting. The USBTMC488 implementation was abandoned during the prototype launch stage because it was not necessary for the correct operation of the module. The SCPI was chosen as the protocol layer, which is a standard for text-based communication with measuring devices using commands. It is a protocol written for interfaces such as RS-232 and, above all, GPIB; therefore, the last specification comes from 1999. The protocol is based on a command tree, where commands can be navigated and individual procedures are called. After selecting the protocol and implementing the USBTMC class, an SCPI command parser was written to define the command tree structure. The most important function in SCPI handling parses the string and returns a pointer to the function that handles the given command, if found. Figure 8 shows a block diagram of the relationship between the various software elements. In multithreaded software, it is important to remember to exclude each other when checking the FPGA so that the threads do not try to use the SPI interface at the same time.

To control and download data from the module, software that would operate the device at the computer level must be written. A basic program was written to download data and prove the correct operation of the device; thus, the graphic interface was abandoned, and the software operation was limited to calling it from the console. The software was written for GNU/Linux systems for everyday use and the programming language was C. Make ran a compilation and linked it to the GCC compiler. The Linux kernel has a built-in driver for devices that use the USBTMC class; therefore, they appear in the system as a file and can perform read and write operations, as with any typical file. Therefore, the operations performed in the software were limited to file operations, except for module support, where unbuffered file operations were used so that the data were transferred without delay. Unbuffered file handling is limited to using file operation functions without the "*f*" prefix, for example, an entry to a file is made by calling *write* () instead of *fwrite* (). The program relied on a sampling configuration file. Successive lines were read from the configuration files. The first line should contain the analog sampling clock divider and the second line should contain the digital sampling clock divider. If any of the divisors are zero, the sampling subsystem is turned off. The third line contained the number of milliseconds during which data were retrieved. The successive lines of the configuration file contain the successive configuration values of the analog inputs. After starting the program, it checked the given configuration file, then started sampling and cyclically downloaded from the module information about the data content in FIFO buffers. After exceeding the appropriate data threshold, they were read and saved in the form of pure binary data in a temporary file. After the sampling time had elapsed, the sampling was turned off, the remaining data in the FIFO were read, then the temporary files were interpreted and, based on the analog input configuration table, converted into appropriate voltage values placed in the output CSV file. The program was called a console with two arguments. The first argument was the path to the acquisition configuration file and the second was the path to the USBTMC device file.

## 3. Results

The first test examined a single analog channel for incremental verification, which facilitated its identification in the event of an error. A single channel with the maximum sampling frequency was selected, and a sinusoidal signal with a frequency of 1 kHz and an amplitude of 10 V was introduced into the channel. The device was verified as positive. The plot agrees with the generator settings and parallel waveform observed with the oscilloscope (Figure 9a). Subsequently, the correctness of channel switching was tested. The test was run at the maximum sampling rate at which any problems due to too long a capture time would appear as a distortion. Two channels were configured as inputs, and sinusoidal waveforms with a frequency of 1 kHz and an amplitude of 2 V were introduced into them. The introduced sinusoids were 180° out-of-phase for better visibility, as shown in Figure 9b. The module then passes the verification test again. No distortions were observed in the charts. Comparing the plot with the single-channel test, the consequences of multiplexing can be observed. Using a single channel, two 1 kHz sine waveforms were used to obtain 1000 samples with a sampling rate of 500 kS/s. When measuring the two channels, the sampling frequency for each channel decreased twice, so that two waveforms of the same frequency took twice as few samples.

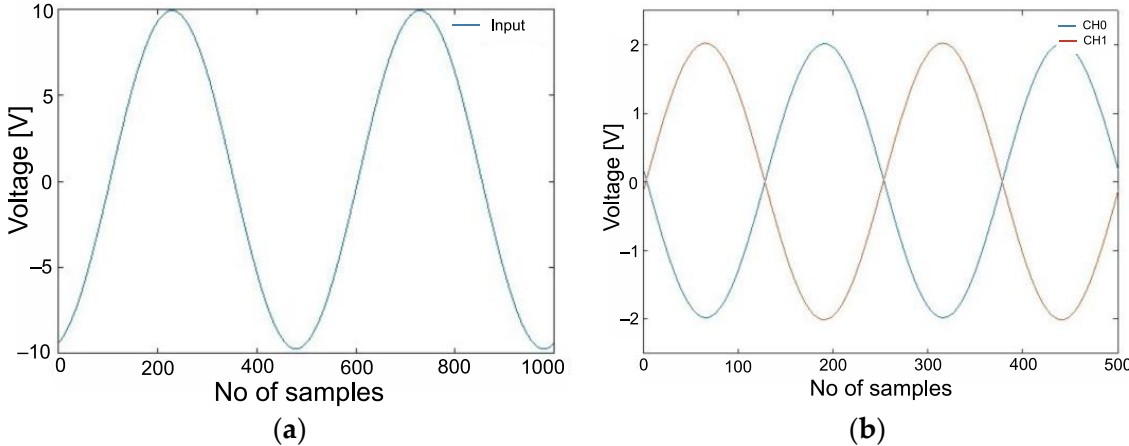

**Figure 9.** Verification of single-channel operation (**a**); verification of channel switching (**b**).

In the next stage of the research, the reinforcement was verified. The channel-switching tests were repeated by changing the amplitude of the waveforms to 1 V and changing the measurement configuration file such that the device amplified the waveform of the first and second channels twice and five times, respectively. Owing to the software calculations on the computer, the output data in the form of CSV should exhibit sine waves with an amplitude of 1 V. The test was repeated for different gain values (two- and five-fold). The same sine wave was measured but with different gains; therefore, the plot should overlap, but the raw data should have clear differences between each consecutive sample and similarities between every second sample (Figure 10a). There was a noticeable difference in the value between the two- and five-fold amplified channels. The correct operation of the differential inputs was then checked. In this study, the measurement of the differential pair was configured, and on both inputs, a triangular wave of 5 V amplitude and 1 kHz frequency shifted in phase by 180° was given. A separate measurement of each signal of the differential pair was added to the configuration to verify the accuracy of the obtained differential signal. The verification results are as expected. Owing to the phase shift of the triangular waveforms, their amplitudes increased and the differential measurement exhibited a triangular waveform with an amplitude of 10 V (Figure 10b). It was decided to repeat the verification using the gain, as earlier verification of the gain did not use differential measurements. The program was set up to obtain a differential sample from the pair of channels 0 and 8 with two-fold gain, as well as from the pair of channels 1 and 9 with

five-fold gain. In addition, the channels constituting the pairs were measured individually. The measured signals were sine and square waves. The signals were connected inversely for each pair, so the resulting waveforms should be in antiphase (Figure 10c).

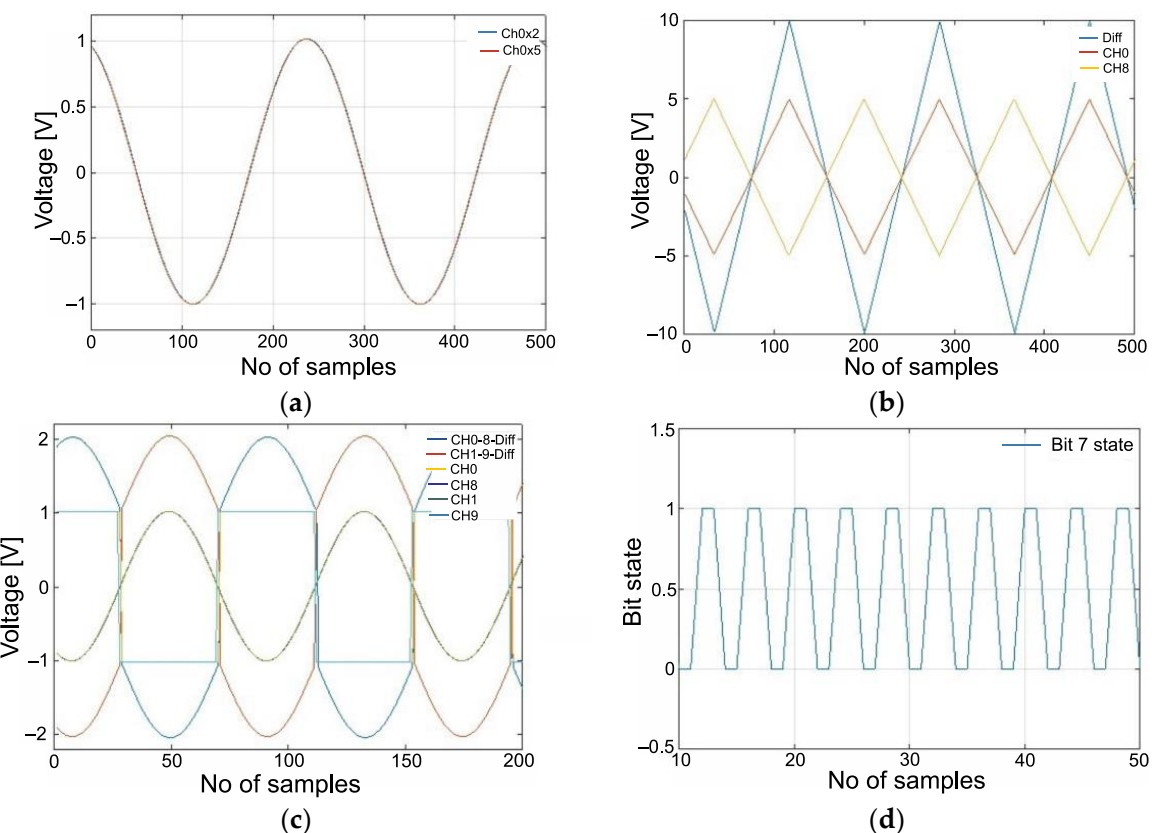

**Figure 10.** Repeated verification of the gain (**a**); verification of the differential measurements (**b**); test of the differential measurement with gain (**c**); state of the digital input during verification (**d**).

To verify the operation of the digital inputs, a square wave with a 50% fill and 125 kHz frequency was introduced on one of the digital inputs, and the predicted waveform at a sampling frequency of 500 kS/s was the change in the input state for every second sample. The digital input behaved as predicted (Figure 10d).

The last element was the measurement of the *CMRR* of the device. Owing to the lack of typical laboratory measuring equipment, it was decided to introduce the same signal to both inputs of one of the differential pairs, and then to test the changes in this input, which in the ideal situation should remain equal to 0. First, a triangular signal with an amplitude of 5 V was introduced, and the test was carried out for a 1 kHz signal (Figure 11a). One can note the representation of the triangular signal in the change in the value of the differential measurement, which indicates that the common signal was not perfectly attenuated. In addition, a zero shift of almost 1 mV was observed. It is most likely due to the fact that the device software at the prototyping stage does not have any form of calibration and error removal such as zero shifts. From this measurement, the common-mode rejection factor was estimated using the ratio of the difference between the maximum and minimum values of the two measurements. A formula was used for this purpose:

$$CMRR = 20 \cdot \log\left(\frac{\Delta V_{diff}}{\Delta V_{input}}\right), dB; \tag{3}$$

where $\Delta V_{diff}$ is the difference between the maximum and minimum values of the differential measurement, $\Delta V_{input}$ is the difference between the maximum and minimum

values of the common signal, and *CMRR* is the estimated value of the common-signal rejection coefficient.

The values from the *CMRR* chart are approximately 74 dB. With a 16-bit converter, this value should be greater than 90 dB to avoid disturbing differential measurements. The measuring amplifier provides a 98 dB *CMRR*, but there are several circuits in front of it. In addition, the module parameter is measured by the module itself, which may be associated with errors such as the DNL of the transducer, which is 0.75 LSB. This means that during the measurement, there could have been an error by LSB; if we decrease $\Delta V_{diff}$ by 1 LSB, the result will be 76 dB.

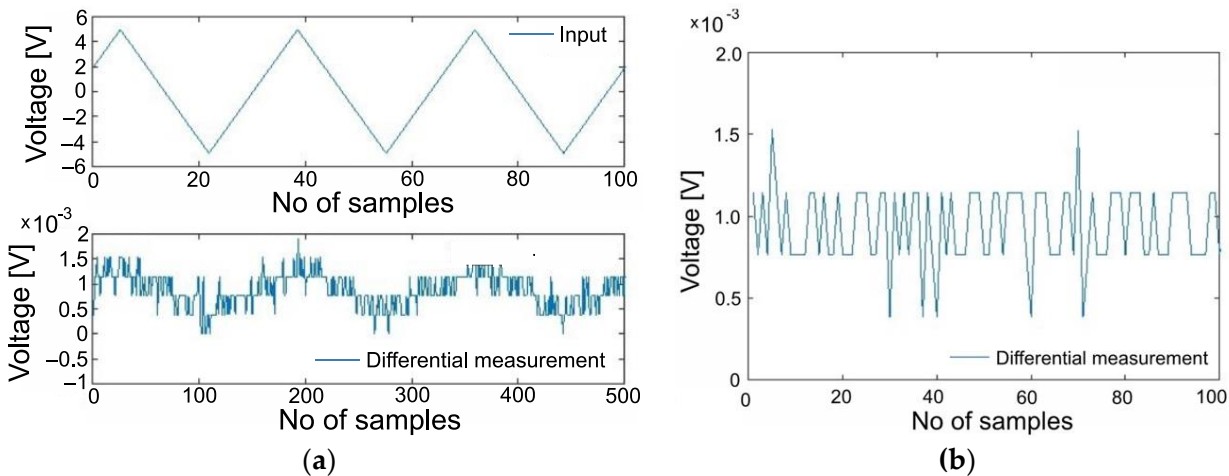

**Figure 11.** CMRR measurement (**a**); noise at the measurement input (**b**).

Owing to the unsatisfactory results of the measurement of the common signal attenuation coefficient, it was decided to measure the noise with the generator connected but displaying 0 V (Figure 11b). Disturbances that reach a peak at 1 mV are clearly visible. The source of interference cannot be unequivocally determined because the device was not tested under laboratory conditions, nor was it powered from a controlled power source, but directly from a computer running the communication software. However, it can be assumed that the cause of the noise may be the interference from the devices located at the test site. In addition, interference could be introduced by converters located on the PCB or digital lines entering the analog block, that is, gain control, channel selection, and ADC interface.

## 4. Discussion

After identifying the topologies used in the analog part of the device, the components were selected and the measurement module was designed, constructed, and programmed. The paper presents the process of data acquisition from the form of an analog voltage signal to the form of a digital signal. In particular, the entire process of signal sampling and quantization as well as digital recording was shown. Our device is capable of acquiring analog signals with voltages in the range of $\pm 10$ V and digital signals in the TTL 5 V standard with a frequency of at least 250 kS/s. The A/D converter used satisfied the resolution requirement of at least 14 bit, and the analog inputs were configured for differential measurements. The device can be controlled by a computer, and the data can be downloaded via the USB interface. Taking into account the versatility and low production costs of the proposed device along with the possibility of acquiring analog signals from the above-mentioned range, the ability to acquire data from a wide range of sensors equipped with analog voltage outputs was obtained.

The design requirements were exceeded because the converter had a resolution of 16 bits and allowed for sampling at a frequency of 500 kS/s. Moreover, correct measurements could be obtained in the voltage range of $\pm 12.5$ V. The possibility of amplifying the

input signal has also been added to maintain a high resolution when measuring signals with amplitudes that are significantly lower than the measuring range. The device operation was verified. A simplified measurement of the common mode attenuation coefficient was made, the value of which was 76 dB; however, to unambiguously measure the parameter and make corrections by increasing its value, laboratory equipment should be used.

Attention was paid to the relatively high input noise of the measurement channels, and the potential causes were identified. This problem can be further investigated and solved by providing an appropriate research environment.

The proposed module is comparable to a commercial card and has the same resolution, but a higher sampling frequency. In addition, it is several times cheaper and has more memory than, for example, a commercial card. The module's *CMRR* parameter is much lower than that of the commercial PCB, but its exact value is unknown because the commercial card has specifications only for common signals of up to 60 Hz. The advantage of the proposed system is the possibility of implementing additional functions in a device.

The device itself can be improved to meet the SCPI standard, and the USBTMC class can be extended with the USBTMC488 subclass, which eliminates the need to constantly query the amount of data in FIFO from the computer while taking the measurement. Additional components of the device can be used to run the tests before designing a prototype for use during operation. In the current prototype, it is possible to add support for the Ethernet interface; adding Wi-Fi or supporting a microSD card is not possible, as the power consumption will exceed 500 mA allowed by the USB 2.0 standard. The device could then be powered by a charger with a higher output current, but this would mean resigning from communication with the computer via USB.

Because the TCM memory of the microcontroller is not used when writing the software, the use of less than half of the RAM and approximately 15% of the flash memory, as well as the core clocking at 192 MHz, with a possible 480 MHz, many functionalities can be added to the microcontroller software, such as resetting the unbalanced voltage in measurements or the segregation of samples, such that each tested channel has its own FIFO.

## 5. Conclusions

The example presented shows that devices constructed in a domestic environment require a laboratory for full characterization. However, they can achieve parameters that are comparable to those of commercial products. Designing analog inputs in a manner that allows them to be configured in pairs to form differential inputs allows for a reduction in the number of components and makes the device more universal. By using components that combine multiple functions, such as a gain-programming amplifier, one can reduce the number of components on a PCB and add functionality to the design without increasing the number of components. Fully differential amplifiers work well as converters of signals into differential pairs. The use of components with integrated passive components adjusted with parameters in the production process allows the achievement of precise amplification and attenuation without the need for calibration in the laboratory. When high-precision switching of digital signals is required, it is worth using FPGA, which is not burdened by instruction execution delays. In addition, FPGAs allow for a fully independent operation of the systems implemented in them.

Choosing an existing USB class when implementing this interface can speed up the work, because drivers for operating systems are already written for the existing classes. The Unix-based system ideology that "everything is a file" allows universal hardware support through the usual file read-and-write operations.

The assumed cost minimization criterion was also satisfied while maintaining the appropriate parameters of our measurement card. Table 5 presents the parameters of a commercial measurement card provided by a recognized manufacturer for this type of assortment and different solutions. As can be seen from the cost position in the table, the price of our device is almost 10 times lower.

**Table 5.** Comparison of the parameters of the designed module with the parameters of the commercial National Instruments measurement card model USB-6211.

| Parameter | Developed Module | USB-6211 |
|---|---|---|
| Number of analog inputs | 16 | 16 |
| Number of inputs in differential | 8 | 8 |
| ADC resolution | 16 bit | 16 bit |
| Sampling frequency | 500 kS/s | 250 kS/s |
| Input measuring ranges | ±1 V; ±2 V; ±5 V; ±10 V | ±1 V; ±2 V; ±5 V; ±10 V |
| CMRR | 76 dB at 1 kHz, requires a more accurate measurement | Above 100 dB up to 60 hZ |
| FIFO size | 1000,000, 2 separates | 4096 |
| Input configuration table length | 64 k | 4095 |
| Number of digital outputs | 8 | 4 |
| Number of digital inputs | 8 | 4 |
| Cost | Cost of all components (also additional) in unit price approx. 650 PLN | 6880 PLN (current price) |

**Author Contributions:** Conceptualization, M.S. and S.P.; methodology, S.P.; software, M.S.; validation, A.B., S.P. and S.G.; formal analysis, S.P.; investigation, A.B.; resources, M.S.; data curation, M.S.; writing—original draft preparation, M.S. and S.P.; writing—review and editing, S.G.; visualization, A.B.; supervision, S.G.; project administration, S.P.; funding acquisition, S.G. All authors have read and agreed to the published version of the manuscript.

**Funding:** This research received no external funding.

**Data Availability Statement:** Data available on request: sebastian.pecolt@tu.koszalin.pl.

**Conflicts of Interest:** The authors declare no conflict of interest.

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
