# Peer review of "Design of a Low-Cost Measurement Module for the Acquisition of Analogue Voltage Signals"

_electronics, doi:10.3390/electronics12030610_

Round 1

Reviewer 1 Report

In this paper, the authors are presenting a Universal multichannel measurement card. I believe there are many issues that the authors should address before this paper can be accepted for publication.

1. The authors tried to do a good job of giving a background of measurement modules in the introduction section. I believe the authors need to explain better what new they are offering through this paper ( page no 2, Lines 49 to 66). 

2. All the figures provided in the paper were of poor resolution except Figure 7. I believe the authors should definitely replace them.

3. The number of references used in this paper is inadequate for a journal publication. 

4. The text in Figure 8. block diagrams are not readable. Since there is no page limitation, I believe the authors should have made it bigger that way text could be readable. 

5. I wish the authors would have compared the measurement results obtained from their novel design with some other well-established or credible instrument. And presented the error between them. That way, they can show the accuracy of their proposed multichannel measurement card.

Author Response

Dear Associate Editor and Reviewer,

The authors would like to thank the anonymous reviewer and the associate editor for their effort in reviewing the manuscript and for their valuable and constructive comments and fruitful observations, which helped in improving the quality of the manuscript to a publishable standard. Detailed below are the responses to the reviewers’ comments and suggestions. Reviewers’ questions and comments are in BLACK and the authors’ answers and comments are in RED.

Reviewer 2 Report

Please, find the attached PDF file. 

Author Response

Dear Associate Editor and Reviewer,

The authors would like to thank the anonymous reviewer and the associate editor for their effort in reviewing the manuscript and for their valuable and constructive comments and fruitful observations, which helped in improving the quality of the manuscript to a publishable standard. The response to the reviewer's comments and suggestions are included in an additional file. Reviewers’ questions and comments are in BLACK and the authors’ answers and comments are in RED.

Round 2

Reviewer 1 Report

I am grateful to all authors for incorporating the suggested changes in the final manuscript. Everything looks better compared to the previous version.

Reviewer 2 Report

In my opinion, the revised version of the paper can be published.